# Long-Term Engraftment of Cryopreserved Human Neurons for In Vivo Disease Modeling in Neurodegenerative Disease

**DOI:** 10.3390/biology14020217

**Published:** 2025-02-19

**Authors:** David J. Marmion, Peter Deng, Benjamin M. Hiller, Rachel L. Lewis, Lisa J. Harms, David L. Cameron, Jan A. Nolta, Jeffrey H. Kordower, Kyle D. Fink, Dustin R. Wakeman

**Affiliations:** 1Department of Neurological Sciences, Rush University Medical Center, Chicago, IL 60612, USA; 2Department of Neurobiology, Barrow Neurological Institute, Phoenix, AZ 85013, USA; 3Stem Cell Program, Department of Neurology, Institute for Regenerative Cures, University of California Davis, Sacramento, CA 95817, USA; pbdeng4@gmail.com (P.D.);; 4FujiFilm Cellular Dynamics Inc., Madison, WI 53711, USA; 5Banner Neurodegenerative Disease Research Center, Arizona State University, Tempe, AZ 85281, USA; 6Department of Psychiatry, Yale University, New Haven, CT 06511, USA

**Keywords:** human cell transplantation, cryopreservation, experimental animal models, immune-deficient models, induced pluripotent stem cells (iPSCs), central nervous system

## Abstract

Scientists are exploring ways to better understand and treat brain diseases by transplanting human neurons into the brains of animals. In this study, researchers tested whether certain human brain cells, made from stem cells and preserved for later use, could survive and work in different animal models, including rats, mice, and monkeys. These cells, which help regulate brain activity, were injected into specific brain regions and monitored over time. The transplanted cells not only survived but also grew connections with the animals’ brain tissues like mature, healthy nerve cells. In one model of Huntington’s disease, a condition that damages brain cells, the transplanted cells survived for six months and even took up harmful proteins from the surrounding diseased brain, providing a new way to study disease processes. The findings also showed that these cells can be safely stored and used reliably in research, making them a promising tool for studying brain diseases and testing new treatments. This work could pave the way for breakthroughs in understanding complex neurological conditions and developing therapies that may one day benefit patients.

## 1. Introduction

Transplantation of human neurons derived from pluripotent stem cells has been investigated extensively for neuronal cell replacement therapy in neurodegenerative diseases. Midbrain dopamine neurons derived from embryonic or induced pluripotent stem cells (iPSCs) offer considerable promise as an alternative source to fetal neurons for the treatment of Parkinson’s disease (PD), and several groups are performing or preparing for first-in-man clinical trials to assess safety in small patient cohorts [1,2]. Likewise, Huntington’s disease (HD) has long been targeted for cell replacement of striatal neurons [3,4,5] with advances being made toward the generation of medium spiny neurons from human pluripotent stem cells [6,7,8,9] or induced neurons, neurons directly generated from somatic cells. The latter have been shown to retain their aged epigenetic signature and recapitulate disease pathology with aggregation of mutant huntingtin protein (mHtt) and associated physiological deficits when derived from HD patients [10,11,12].

While grafted neurons derived from human stem cells have been widely reported to restore functional deficits in animal models of neurodegenerative diseases including PD, HD, Alzheimer’s disease, and Amyotrophic Lateral Sclerosis [13], far less is known about how the diseased central nervous system (CNS) impacts the functionality of grafted human neurons. Understanding this dynamic is integral for long-term clinical success. All of these neurodegenerative diseases share a common aggregated protein pathology in the CNS when interrogated in post-mortem patient tissues. Extensive evidence from numerous reports across these disorders suggests that cell-to-cell transport of pathological protein species may be responsible for at least some portion of transmission throughout the neuraxis [14].

Clinical evidence for the spread of alpha-synuclein pathology from PD patients’ brains to grafted fetal dopamine neurons has been documented in several case reports [15,16]. Likewise, in reports of HD in vitro studies [17,18], transgenic HD animal models [18,19,20,21], and HD patients grafted with fetal neurons [22,23] have repeatedly demonstrated evidence for the transmission of mHtt disease pathology between host and donor cells. Furthermore, a report demonstrated the transmission of Alzheimer’s disease pathology from transgenic mice into grafted human cortical precursor cells, resulting in pathological hallmarks of the disease, including amyloid plaque deposition, hyperphosphorylated tau, and dystrophic neurites, as well as downregulation of genes associated with cognition and synaptic transmission [24]. Similarly, a combination of preformed alpha-synuclein fibrils and AAV-alpha synuclein were recently shown to seed pathology in dopaminergic neuron transplants when co-seeded in the striatum of immunosuppressed Sprague–Dawley rats [25]. These collective results provide evidence that grafted human neurons can be used to model disease transmission and possibly test the efficacy of new experimental therapeutics.

Xenotransplantation of human neurons into non-diseased healthy animals has also been utilized as a novel experimental model to study development. Humanized rodent brain chimeras using cortical pyramidal neurons were shown to follow a prolonged human-like maturation timeline and functionally integrate with mature synapses in the mouse cortex though displayed host-species intrinsic electrophysiologic properties [26].

The neurodegeneration field strongly lacks relevant test systems that can assess how toxic protein species damage human brain systems across long time spans. Animal models that live long enough to assess human xenografts and faithfully emulate disease-relevant pathophysiology are few and far between. Furthermore, there are relatively few reproducibly well-characterized human neurons of a known lineage that have been validated by numerous biochemical and functional assays across production lots available to the public for mass investigation at scale. Such tools are pertinent for systematic, disease-specific drug screening.

The cryopreservation of post-mitotic GABAergic-like neurons derived from iPSCs (iGABAs) is a major advantage in studying graft survival and neurite outgrowth in translational models of disease as they can be reliably and reproducibly thawed, providing rapid access to large numbers of highly characterized cells [27,28]. Conventionally, iPSC-derived neurons are prepared fresh from culture and must immediately undergo a rigorous set of quality control release criteria prior to use. This laborious process is difficult to synchronize at scale and requires expert technical support to culture and prepare the cells, often resulting in variation in batches of cells that do not pass release criteria and thus are not amenable to large-scale experiments. Cryopreservation circumvents many of these technical hurdles as it allows for the banking of large production lots which can then undergo both functional and safety screening procedures prior to use. It also allows for the final validated cell product to be shipped and stored providing immediate access when needed. Batch analysis of iGABA commercial production lots consistently demonstrates stability of critical quality attribute release criteria up to 5 years post-cryopreservation, the longest time point assayed by the manufacturer.

While the utility of human iPSC-derived neurons as an in vitro diagnostic tool to interrogate mechanisms of human disease has been widely published [29], the in vivo use as a surrogate to investigate pathological protein transfer in the brain is less studied. To address this critical gap, we examined the potential of cryopreserved iGABAs to engraft into the rodent and nonhuman primate brain and describe the development of in vivo test systems using a well-characterized iPSC-derived human neuronal cell line to study and access early seeding events relevant to proteinopathies of CNS. We demonstrate long-term survival and innervation of these cells in the diseased and aged brain using three translationally relevant species (mouse, rat, monkey) including a novel immune-deficient transgenic HD mouse model, in which grafted cells were associated with host-derived aggregated mHtt six months post-transplantation.

## 2. Materials and Methods

### 2.1. Single Cell Sorting

iGABA neurons were thawed according to the manufacturer’s user guide, centrifuged at 400× *g* for 5 min, and resuspended at ~2 × 10^6^ cells/mL in sort buffer consisting of PBS (Gibco, Waltham, MA, USA), 1 mM EDTA (Invitrogen, Waltham, MA, USA), 0.5% BSA (Fisher, Waltham, MA, USA), and 25 mM Hepes (Gibco). Single cells were sorted into 96-well PCR plates containing 5 µL per well of lysis buffer consisting of PCR-certified water (Teknova, Hollister, CA, USA), 5X VILO reaction mix (Invitrogen), 20 U/µL SUPERase•In (Invitrogen), and 10% NP-40 (ThermoFisher). Cells were sorted using a SONY SH800 Cell Sorter (Tokyo, Japan) and software using a 100 µM sorting chip and 4 PSI. Prior to sorting, gates were set according to size, granularity, and doublet exclusion. 

### 2.2. Quantitative PCR

Single-cell gene expression consisted of a two-step RT-PCR followed by qPCR performed with the Fluidigm BioMark HD system (San Francisco, CA, USA). All steps were executed as described in Fluidigm’s Real-Time PCR user guide (68000088), appendix C. Briefly, cells were sorted into PCR plates as described above and stored at −80 °C until use. Plates were thawed on ice, incubated at 65 °C for 90 s then chilled on ice for 5 min. Superscript Vilo (Invitrogen) and T4 Gene 32 Protein (New England BioLabs, Ipswich, MA, USA) were added directly to the PCR plate and then incubated for 5 min at 25 °C, 30 min at 50 °C, 25 min at 55 °C, 5 min at 60 °C, then 10 min at 70 °C. The preamplification was performed with the TaqMan PreAmp Master Mix (Thermo Fisher Scientific), pooled 48 primer pairs (50 nM final concentration each primer) and EDTA (4 mM final concentration) for 20 cycles (5 s at 96 °C, 4 min at 60 °C). Unincorporated primers were removed by Exonuclease digestion for 30 min at 37 °C followed by 15 min inactivation at 80 °C (New England BioLabs). Samples were diluted 5-fold with DNA suspension buffer (Teknova). The qPCR was performed with Fluidigm’s 48.48 dynamic array IFC on the Biomark HD using 2X SsoFast EvaGreen Supermix with Low Rox (Bio-Rad Laboratories), 20X DNA Binding Dye Sample Loading Reagent (Fluidigm), and primers at a final concentration of 500 nM. The cycling parameters were as follows: hot start at 95 °C for 60 s, then 30 cycles of 96 °C for 5 s, 60 °C for 20 s, then 60–90 °C melting curve. Primers (Table 1) were designed by Fluidigm’s Delta Gene service. Heatmap analysis was performed using the “Singular Analysis Toolset” in R.

### 2.3. Transplantation of iGABAs

iCell Neurons (iGABAs) were thawed and injected bilaterally into the striatum of (n = 3 5-month old Sprague–Dawley and (n = 6 11-month old athymic-RNU NUDE rats (4.5 × 10^5^ cells/hemisphere; 1 injection/hemisphere; 1.5 × 10^5^ cells/μL × 3 μL); unilaterally into (n = 15) 3–4-month-old transgenic R6/NSG and (n = 2) WT/NSG littermate mice (1.4–1.6 × 10^5^ cells/hemisphere; 1 injection/hemisphere; 7.0–8.0 × 10^4^ cells in 2 μL); unilaterally into (n = 2) adult Cynomolgus macaque monkeys (3.75 × 10^6^ cells/hemisphere; 3 injections/hemisphere; 1.5 × 10^5^ cells/μL) using standard stereotaxic procedures as previously described [28,30,31,32]. Stereotaxic coordinates for rats: AP: 0.4 mm, ML +/−3.3 mm, DV −5.2 mm. Stereotaxic coordinates for mouse: Anterior–Posterior: 0.5 mm; Medial–Lateral; −1.7 mm; Dorsal–Ventral −2.5 mm. Coordinates for the Cynomolgus monkey striatum were determined based on MRI guidance using a Stealth Navigation Unit (Medtronic, Dublin, Ireland). The monkeys received three deposits of cells into the rostral (10 μL), middle (10 μL), and caudal (5 μL) striatum. Immunosuppression of Sprague–Dawley rats (cyclosporine) and Cynomolgus monkeys (tacrolimus, mycophenolate mofetil, and prednisone) was performed as previously described [28,32]. The mice, rats, and monkeys were humanely euthanized according to the schedule in Table 2 as previously described [28,30,31,32]. All in vivo procedures were performed with IACUC approval from Rush University and/or the University of Illinois at Chicago and the University of California at Davis.

### 2.4. Immunohistochemistry and Stereology

For rat and monkey tissues, immunohistochemistry and unbiased stereological analyses were performed as previously described [28,30]. Parameters for counting HuNuclei were as follows: Counting Frame 50 μm × 50 μm and Grid Size 125 μm × 125 μm for a total area counted of 16% with an average coefficient of error of 0.08 (Gundersen, La Crosse, WI, USA, m = 1) using Stereo Investigator v10.40 (MBF Biosciences, Williston, VT, USA). A list of antibodies and staining specifics can be found in Table 3. For mouse tissues, fluorescent images were captured using an Observer Z1 microscope with a motorized stage (Zeiss, Oberkochen, Germany). Whole brain imaging was performed with a 20×/0.8 numerical aperture—apochromatic objective and stitched together using ZEN 2.6. High magnification images were taken with a 40×/0.95 numerical aperture—apochromatic objective and inserted ApoTome slider (Zeiss).

## 3. Results

### 3.1. In Vitro Characterization of iGABAs

Cryopreserved iGABAs derived from human blood samples via episomal reprogramming and forebrain patterning that passed the same release criteria were acquired from multiple production lots and thawed fresh for all experiments (FujiFilm Cellular Dynamics Inc., Madison, WI, USA). iGABAs retained high viability with gene and protein expression profiles consistent with predominantly the forebrain GABAergic neuron lineage in vitro [27,28,33]. Single cell gene expression arrays for 30 neuronal-fate genes were performed across 7 independent lots of iGABA cells with a total of 613 cells analyzed (Figure 1). The experiments conducted here were consistent with other reports using iGABA cells. Experiments performed by the cell manufacturer and other end users have consistently demonstrated that iGABAs are a heterogeneous combination of ~80–90% GABAergic neurons, ~10–15% glutamatergic in composition with a small population (<5%) of pre-mitotic replication competent progenitors. iGABAs have been shown to generate Nav, Cav, and Kv voltage gated-currents and respond to GABA ligands in patch clamp experiments [33] and mature when assayed for electrophysiological parameters including repetitively firing in culture, changes in major voltage currents, and action potential duration consistent with neuronal lineage [27]. iGABAs grown in culture on numerous substrates over extended periods have been shown to extend neurites as well as express MAP2 (>95%) and the lower cortical layer pyramidal neuron marker CTIP2 (44%) in the absence of SATB2 (upper cortical layer marker) or GFAP (astrocyte) expression up to 28 days in vitro [27].

### 3.2. Survival and Outgrowth of iGABAs in Immunosuppressed Rat Brain

We utilized a previously developed method to thaw cryopreserved neurons for transplantation into animal models [28]. To determine engraftment potential in the CNS, iGABAs were prepared without additional subculturing and injected bilaterally into the striatum of Cyclosporine-A immunosuppressed Sprague–Dawley rats. Three months following injection, human-specific Neural Cell Adhesion Molecule+ (HuNCAM+) grafts were located at the injection sites with extensive fibers projecting into host parenchyma and along white matter tracts. Fiber innervation extended well beyond the striatal zone of injection (Figure 2A). Processes were found innervating most rostrally throughout the genu and forceps minor corpus callosum, cingulate cortex area 1, M1 motor, M2 motor, and frontal cortex area 3 (Figure 2B). Caudally, extensive fiber innervation was noted dorsally through the cingulum and hippocampus, as well as more ventrally predominantly coursing through the areas of the internal capsule, thalamus, entopeduncular nucleus, amygdaloid nucleus, nucleus of Meynert, zona inserta, medial forebrain bundle, and nigrostriatal bundle (Figure 2C), which extended into the midbrain, innervating substantia nigra, red nucleus, superior colliculus, and periaqueductal gray area (as far caudal as was anatomically assessed) (Figure 2D,E). These data demonstrate proof of concept that cryopreserved iGABAs survive and continue to mature for up to three months following engraftment into the immunosuppressed rodent brain.

### 3.3. Long-Term Engraftment and Biodistribution in Aged Immunodeficient Rat Brain

Immunodeficient animals offer a unique opportunity to investigate xenograft–host interactions without the confounds of immunosuppression drugs that can have toxic side effects when administered over long periods of time [34]. We utilized the adult-aged (11-month-old) athymic RNU-Nude rat as a means to test the long-term survival and brain distribution of transplanted iGABAs. Similar to our results in immunosuppressed Sprague–Dawley rats, iGABAs engrafted (Appendix A) and projected neuronal HuNCAM+ fibers deep into the striatum and along the corpus callosum white matter tracts short-term (1-week; Appendix A), demonstrating that human neurons can integrate into the adult immunodeficient rodent brain.

At 9 months post-transplantation (n = 5; Figure 3, Appendix A), the majority of transplanted cell bodies remained in the striatum near the site of injection, bilaterally in both hemispheres, although some human-specific nuclei+ (HuNuclei+) donor cells were found to have migrated along the corpus callosum and external capsule while some small pockets of cells were localized to the cortex, likely due to needle reflux at time of injection. Similar to our previously published results using cryopreserved dopamine neurons [28], cell viability was maintained long-term (123,205.8 +/− 17,528.7 HuNuclei+ cells/hemisphere, n = 10 hemispheres; mean +/− SEM; Figure 3A–C; Table 4). The finding of a few migratory cells along white matter tracks (Appendix A) is consistent with the small portion of replication-competent precursor cells within the iGABA population. In addition, few HuNuclei+ cells co-localized with cell proliferation marker Ki-67, though no tumors or tumor-like structures were found, indicating that any replication-competent progenitor cells were not continuing to divide and proliferate unchecked (data not shown), consistent with pre-clinical dopamine neuron grafting experiments performed by numerous groups.

Remarkably, by 9 months, iGABAs had matured and innervated rostrally and caudally throughout nearly the entire neuraxis (Figure 3D). HuNCAM+ fibers were located medial to the graft as well as rostrally within motor and orbital cortices and forebrain somatosensory cortical structures. The more dorsal cortical regions were less innervated. Graft-derived fibers were seen preferentially innervating the forebrain striatum and projecting ventro-caudally throughout the striatum and along white matter tracts. Surprisingly, HuNCAM+ fibers were detected well beyond the injection site into thalamic and amygdaloid nuclei and coursing through the midbrain as far as the cerebellum, which is as far caudal as we analyzed. Unfortunately, the spinal cord was not collected to determine if the predominantly GABAergic neuron population projected further into the physiologic ventro-caudal hindbrain and spinal target structures.

Grafted cells were next analyzed for mature forebrain lineage neuronal markers to determine if phenotypic change occurred following long-term engraftment. Many HuNuclei+ cells with large cell bodies colocalized with DARPP-32 (Figure 3E–G) and Calbindin (Figure 3H–J), indicative of striatal medium spiny neurons, suggesting that transplanted cells continued to mature following ectopic transplantation. Future studies will encompass a more extensive analysis of neuronal lineages and address how cell composition affects fiber innervation in discrete brain structures. Taken together, the biodistribution data clearly demonstrate that xenografted human forebrain lineage iGABAs mature, project fibers, and extensively innervate long-distance physiologic targets within the aged (20-month-old) immunodeficient rat brain.

### 3.4. Engraftment of iGABAs in Non-Human Primate Brain

To determine if iGABAs were capable of surviving in the non-human primate brain, cryopreserved cells from the same production lot of previously described rodent experiments were prepared in an identical fashion and transplanted into the striatum of two Cynomolgus monkeys. The brains were analyzed at 1-month post-injection for human lineage markers. Human Cytoplasm+ (HuCyto) grafts were located within the zones of injections (Figure 4A), projecting fibers innervating the host putamen (Figure 4B). These data provide evidence for the ability of iGABAs to survive and innervate the nonhuman primate brain in a similar fashion as in rodents.

### 3.5. Long-Term Engraftment, Biodistribution, and Pathogenic Assessment in Immunodeficient Transgenic HD Mouse Brain

To determine if iGABAs could survive following transplantation into the degenerative brain, cryopreserved cells from the same production lot as the 9-month NUDE rat experiment were prepared in an identical fashion and transplanted into the striatum of a novel immune-deficient transgenic HD mouse. The novel mouse strain was derived from blastocyst injection of the toxic exon 1 fragment used to create the R6/2 mouse model [35] in an immunodeficient NOD.SCID.IL2G (NSG) background. These mice, termed “R6/NSG”, harbor exon 1 of mutant huntingtin carrying approximately 150 CAG repeats, express mutant huntingtin protein and display mutant huntingtin aggregates in neurons. R6/NSG and Wildtype NSG (WT/NSG) littermates were transplanted with iGABAs at 3–4 months of age and brains were analyzed at 6 months post-injection. At 6 months post-grafting, Human Cytoplasm-positive (HuCyto+) transplants projected fibers into the periphery in both R6/NSG and WT/NSG mice (Figure 5). Similar to the rat brain, HuCyto+ fibers projected into the white matter of the ipsilateral corpus callosum and crossed over to the contralateral hemisphere in the R6/NSG and WT/NSG mouse (Figure 5A–E). HuCyto+ fibers projected laterally into the surrounding parenchymal tissue of the striatum in both R6/NSG littermates (Figure 5F–H) and WT/NSG littermates (Figure 5I–K) 6 months post-injection. These data indicate that iGABAs engraft for up to 6 months in a pathogenic HD model and project their fibers in a similar fashion to previously described rat experiments.

### 3.6. Transfer of Endogenous mHtt into Grafted iGABAs

To examine the nuclear transfer of mHtt aggregates, serial sections containing the grafted iGABAs were visualized using an antibody that detects expanded polyQ repeats (MW1). Graft sites were identified as morphologically distinct regions defined by the cytoarchitecture of the graft in reference to the adjacent section labeled with HuCyto. Distinct aggregation was observed within transplanted human iGABAs of R6/NSG mice (Figure 6A,B), but not within the transplant of WT/NSG mice (Figure 6C,D). This finding is suggestive of a potential transfer of mutant Htt aggregates from the host to the transplant, replicating what had previously been observed following fetal tissue transplant in Huntington’s disease patients [22,23].

## 4. Discussion

The utilization of human neurons for disease modeling in vivo offers an unprecedented opportunity for mechanistic investigation of pathological transmission from both the diseased host microenvironment of transgenic animals into transplanted healthy human neurons as well as from disease-bearing patient-derived neurons grafted into the healthy host parenchyma. Given that it is not feasible to test early-stage pre-clinical candidate therapeutics in human patients, “humanized” or “chimeric” animals with functionally innervated human neuronal xenografts represent a powerful new model to rationally advance the therapeutic pipeline for neurodegenerative disease [24,25,26,36]. We examined the potential of well-characterized cryopreserved human neurons to engraft into three different translationally relevant species (mouse, rat, monkey), including the aged rodent brain and a novel immune deficient transgenic HD mouse model, which can be used to assess aggregate-seeding events.

The results presented here strongly support the feasibility of human xenograft rodent models and lay the foundation for future work using similar models as an in vivo experimental tool to interrogate mechanisms of pathogenic transmission as well as the pharmacodynamic effect of new therapeutic drugs for proteinopathies [37,38]. Of note, aged NUDE rats (11 months old) underwent a 9-month period with grafted cells to mimic maturation within the aging brain, reminiscent of HD, PD, and Alzheimer’s disease. Remarkably, iGABAs innervated throughout the entire aged neuraxis that was surveyed at 20 months of age, projecting rostrally into the prefrontal motor cortex and caudally into the midbrain. The fiber innervation pattern was most pronounced in dorsal regions of the brain running rostro-caudally, in contrast to dopaminergic neuron grafts derived from pluripotent stem cells that preferentially innervate their physiologically appropriate targets when placed homotopically or ectopically (A9 or A10 depending on cell patterning) [28,39,40,41,42].

This long-term assessment in the aged brain is consistent with shorter-term experiments using forebrain patterned neural progenitors in the lesioned (6-OHDA or quinolinic acid) or healthy intact striatum of young adult rodents. Doerr et al., 2016 [36] used 8-week Rag2−/− non-lesioned mice; Adler et al., 2019 [41] used 225–250 g female Sprague–Dawley and athymic NUDE rats (young), both intact and 6-OHDA lesioned; Besusso et al., 2020 [9], used 200–250 g male (young) quinolinic acid lesioned athymic NUDE rats). Notably, the innervation patterns described in our work corroborate the early evidence of recapitulation of striato-nigral circuitry in younger quinolinic acid lesioned athymic NUDE rats reported by Bessuso and colleagues [9]. The comparative density of innervating fibers was likely different due to the use of a more mature mixed GABAergic neuronal population in our studies compared to the striatal lineage specified progenitor-like cells used by Besusso and colleagues [9] that have a more defined developmental potential and regenerative capacity. Similarly, we also found extensive subcortical innervation of the motor and somatosensory cortex as thoroughly described in immunodeficient mice grafted with forebrain progenitors [36]. Collectively, these results suggest that graft fiber innervation to anatomically correct targets is being regulated by cell-intrinsic factors.

Historically, animal models of HD utilized toxins like quinolinic acid or 3-nitroproprionic acid [43] to ablate striatal medium spiny neurons and induce functional deficits. While these models generate functional deficits, they fail to recapitulate the chronic nature of the disease or the concomitant proteinopathy and degenerative neuropathology and thus are unfit for mechanistic studies exploring disease progression or disease-modifying therapeutics. To address these shortcomings, the HD field has created numerous transgenic models in which to not only study disease progression but also screen novel therapeutics, ranging from worms, fruit flies, mice, rats, pigs, sheep, and monkeys [44]. A major focus of these efforts has been generating transgenic mouse models by inserting either human or mouse repeats with yeast or bacterial artificial chromosomes or with traditional genome “knock-in” strategies. These models have been created with a range of CAG expansions, and each model system partially recapitulates an aspect of the disease. Transgenic animals that express the full-length human mHtt, such as the BACHD [45] and YAC128 [46], provide the opportunity to study the human protein and aggregation in an in vivo neuronal system but traditionally have slow disease onset and the level of protein aggregation may be mild. The R6/2 mouse model that expresses the human mHtt Exon 1 displays rapid disease onset and robust protein aggregation but is of limited use due to its relatively short lifespan. Until now these models have been limited in assessing human-derived cellular therapies without the need for complex immune suppression strategies that do not allow for long-term assessments [31].

To this end, we tested the hypothesis that cryopreserved human forebrain neurons engraft in multiple species, integrate within the immune-deficient HD mouse brain, and project fibers to anatomically relevant structures in a lineage-specific context. The R6/NSG represents one of the first model systems that express the human exon 1 mutant HTT and enables the long-term survival of grafted human neurons. This model system utilizes the well-studied R6/2 exon 1 model that displays the rapid accumulation of mHtt aggregates in the brain. Historically, in an immune-competent R6/2, the lifespan of the transgenic mouse is only 12–14 weeks [35]. However, in the absence of an active immune system, we have observed R6/NSG mice living for up to 14 months (JAN & KDF unpublished data). A full characterization of this strain, including longitudinal behavioral analysis and characterization following “humanization” with human umbilical cord blood-derived CD34+ is in preparation. This novel xeno-tolerant model allows human-derived cellular products to be evaluated in the presence of human mHtt with the “humanized” immune system [47], creating a model that may more accurately predict the efficacy of a cellular product for HD.

As was hypothesized based on fetal cell grafts in HD patients [22,23], transplantation of healthy neurons into the neurodegenerative environment resulted in the presence and accumulation of mHtt in the engraftment zone, though extracellular mHtt may be related to cell death. These findings are consistent with several studies examining the transfer of the mHtt protein derived from HD patient brains or pathogenic polyQ fibrils administered to WT and transgenic HD mice and normal Cynomolgus monkeys [18,20]. Furthermore, synergistic data supporting transfer of mHtt between grafted cells and the host microenvironment has been demonstrated using both fibroblasts and iPSC derived from HD patients [21]. However, the findings in this study are novel and of significant importance due to the utilization of “xeno-tolerant” rat and HD mouse models. In this context, engraftment and the progressive transfer of mHtt into transplanted human neurons can be observed without the complex and confounding effects that immunosuppression may have on graft survival and outgrowth [48].

In addition, microglia-mediated synaptic pruning, which is integral to normal neuronal development [49], remains intact in NUDE rats and NSG mice as they still have immune cells in circulation but lack the appropriate mature T-cell signaling cascade responsible for graft rejection. Given the possible roles for microglia in mediating synuclein, amyloid beta, and tau toxicity [50,51], the generation of other similar transgenic animal models bearing proteinopathies crossed onto the NUDE and NSG backgrounds would be powerful tools for the interrogation of Alzheimer’s and PD processes. Furthermore, the relative long-term utility of these models will allow for a more faithful in vitro recapitulation [52] of these age-related proteinopathies and, subsequently, greater insights into the mechanisms of action of different isoforms of pathogenic proteins, including mHtt, alpha-synuclein, and Tau.

The ethical dilemmas surrounding human neural xenografts following long-term transplantation are multifaceted, touching on questions of identity, consciousness, and moral responsibility. One major concern is the potential for these transplants to integrate functionally into host brains in ways that could alter cognition, identity, or even subjective experience. According to the National Academies report on human neural organoids, transplants, and chimeras, such research raises the possibility of transplanted human neural tissue influencing host behavior, cognition, or self-awareness, particularly when integrated into nonhuman primates [53]. This concern is amplified when considering the long-term survival and adaptation of human neural cells in host brains, where they might mature beyond current expectations, leading to unpredictable cognitive or moral implications.

A related ethical challenge is the question of moral status. The presence of human neural tissue in another organism or in a significantly altered human brain challenges traditional frameworks used to determine moral worth and rights. Jeziorski and colleagues [54] discuss similar concerns in the context of brain organoids, particularly regarding the possibility of such entities developing forms of consciousness that warrant ethical consideration. If human neural xenografts contribute to the formation of complex cognitive functions or subjective experiences, researchers and ethicists may need to reassess existing guidelines for their use, care, and potential termination. The long-term implications of such transplants necessitate rigorous oversight, public engagement, and ethical reflection to ensure that advances in neuroscience do not unintentionally create morally ambiguous entities or violate fundamental principles of autonomy and dignity.

## 5. Conclusions

This study successfully demonstrates the robust and reproducible engraftment of cryopreserved, well-characterized human forebrain GABAergic neurons across multiple translationally relevant species, including aged and neurodegenerative mammalian models. The long-term survival and extensive integration of these human neurons within diverse central nervous system environments validate “humanized” xenograft animal models as powerful platforms for elucidating the mechanisms underlying neurodegenerative diseases and protein aggregation phenomena.

The ability of iGABAs to project extensively throughout the host CNS, mature phenotypically, and engage with host neuronal networks highlights their potential as an advanced in vivo model for studying human neurodegenerative diseases. Moreover, the transfer of mutant huntingtin protein into grafted neurons within the Huntington’s disease (HD) mouse model underscores the relevance of this system for investigating mechanisms of pathological protein propagation.

These findings establish cryopreserved iGABAs as a reproducible and scalable tool for creating humanized CNS models capable of recapitulating key aspects of disease pathology. By leveraging these models, researchers can elucidate mechanisms underlying neurodegenerative disease progression, assess early pathological events, and evaluate the efficacy of therapeutic interventions in a context that closely mimics human physiology.

Future directions include extending these models to other neurodegenerative diseases, further characterizing graft–host interactions over long durations, and exploring therapeutic strategies aimed at modulating pathological protein transfer. The generation of immune-deficient, proteinopathy-specific transgenic animal models incorporating humanized CNS grafts offers a powerful avenue for accelerating therapeutic discovery and advancing our understanding of fundamental human neurobiology.

## Figures and Tables

**Figure 1 biology-14-00217-f001:**
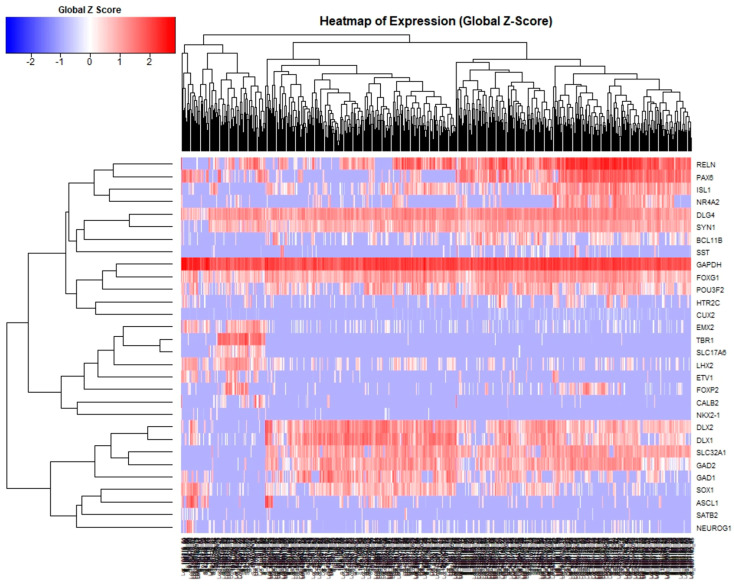
Single cell gene expression array of human iGABA neurons. 30 neuronal-fate genes were analyzed across 7 independent lots of iGABA neurons. Single cell analysis revealed that across multiple lots of the cryopreserved iGABA that expression data was consistent across each lot. Data suggests that the iGABA represents a heterogeneous cell population highly enriched for GABAergic neurons.

**Figure 2 biology-14-00217-f002:**
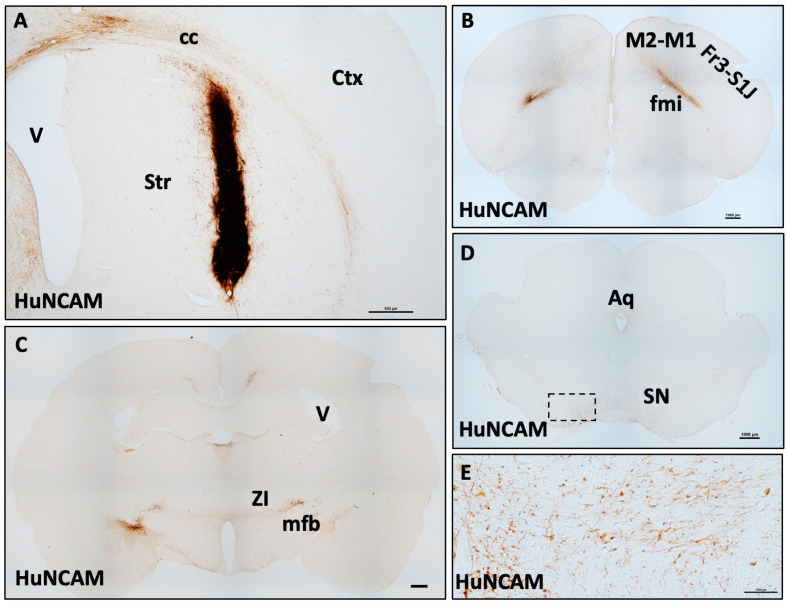
Human iGABAs innervate the immunosuppressed rodent brain. (**A**–**E**). Striatal HuNCAM+ iGABA grafts in immunosuppressed Sprague–Dawley rat at 3 months post-injection innervate deep brain structures from (**A**) the injection site rostrally to the (**B**) frontal cortex and (**C**) caudally all the way to the (**D**,**E**) midbrain. Scales, (**A**) = 500 μm (**B**–**E**) = 1000 μm. Abv: Aq = aqueduct, cc = corpus callosum, Ctx = cortex, Fr3 = frontal cortex, area 3, M1—primary motor cortex, M2 = secondary motor cortex, mfb = medial forebrain bundle, S1J = primary somatosensory jaw, SN = substantia nigra, Str = striatum, V = ventricle, ZI = zona inserta.

**Figure 3 biology-14-00217-f003:**
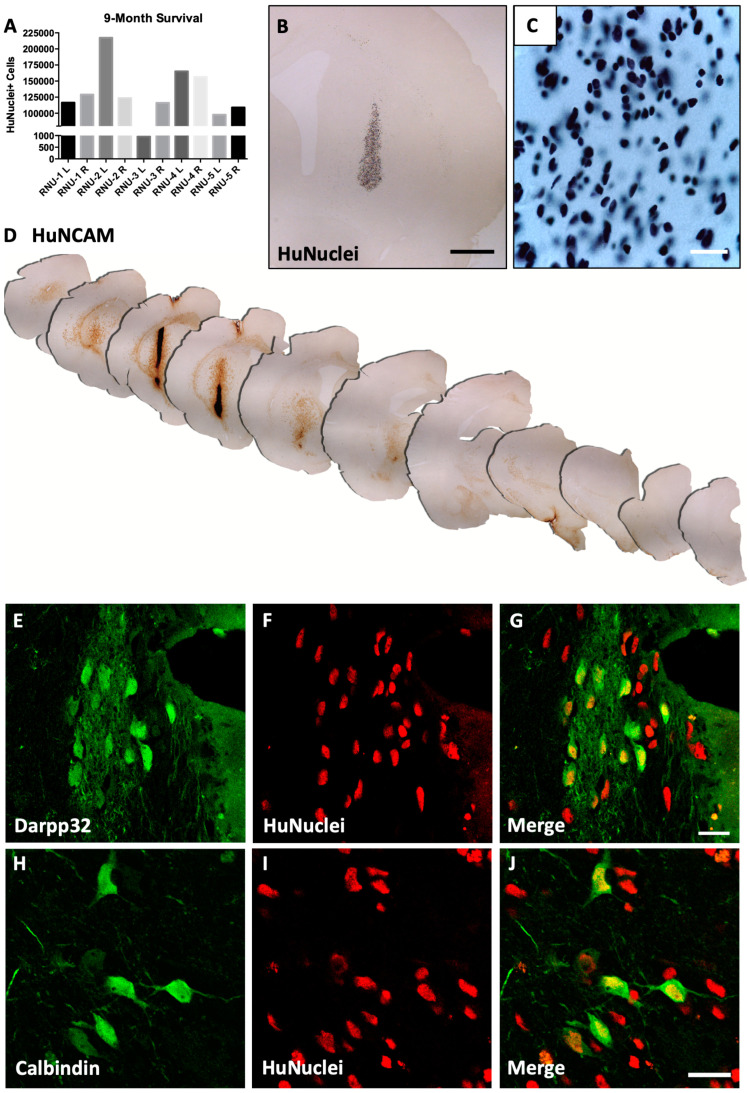
Widespread innervation and maturation of human iGABAs 9-month post-transplantation in RNU-NUDE rat brain. (**A**) Stereological quantification of cell survival (total human-nuclei+ cell). (**B**) Distribution of transplanted human-nuclei+ cells in the striatum and corpus callosum. Scale = 1000 μm. (**C**) High magnification of human-nuclei+ cells within the striatal graft core cell deposit. Scale = 25 μm. (**D**) Human NCAM+ iGABA grafts course the entire neuraxis at 9 months post-injection in the aged immunodeficient rat. (**E**–**G**) Engrafted mature human (Human Nuclei, red) express medium spiny neuron marker Darpp32 (green). Scale = 20 μm. (**H**–**J**) Engrafted mature human (human nuclei, red) express striatal neuron marker Calbindin (green). Scale = 20 μm.

**Figure 4 biology-14-00217-f004:**
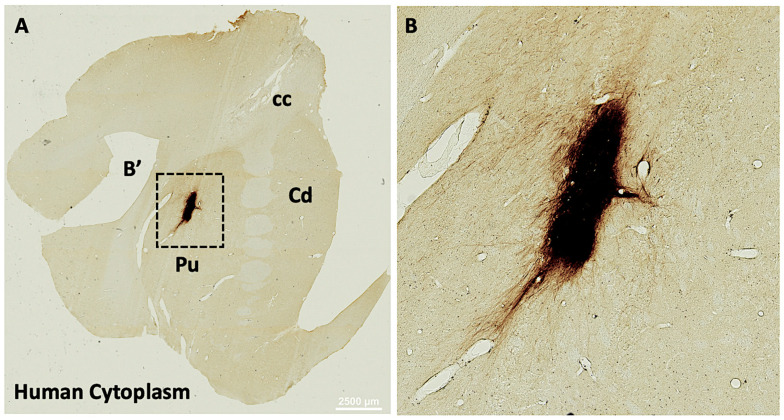
Short-term survival of human iGABAs in Cynomolgus monkey striatum. (**A**) Human cytoplasm+ grafts innervate the nonhuman primate striatum at 1 month post-administration. Scale = 2500 μm. (**B**) Higher magnification of human cytoplasm+ graft from panel g. Abv: cc = corpus callosum, Cd = caudate, Pu = putamen.

**Figure 5 biology-14-00217-f005:**
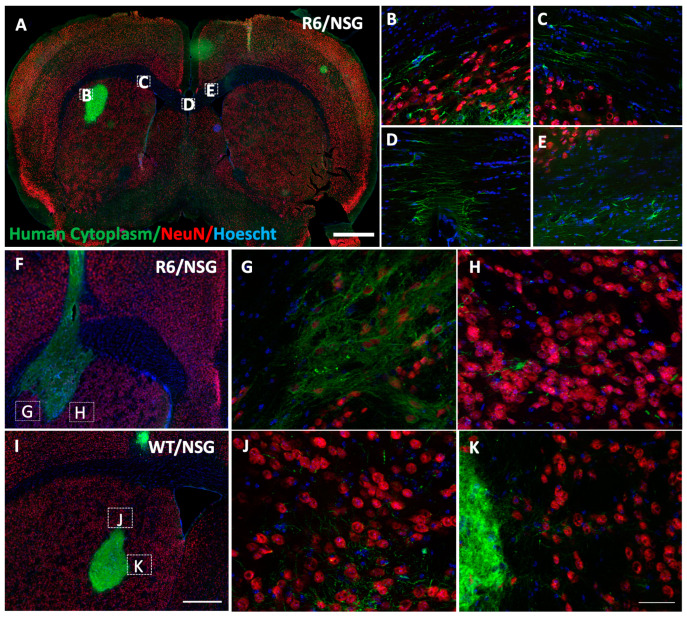
Human iGABAs engraft for up to 6 months in immunodeficient R6/NSG Huntington’s Disease mouse. (**A**) Survival of human-specific human cytoplasm+ iGABAs (green) in the immune-deficient R6/NSG Huntington’s disease mouse brain at the site of injection six months following unilateral injection into the striatum. NeuN (red), Hoescht/DNA (blue); Scale = 1000 μm. (**B**–**E**) Human cytoplasm+ (green) graft fibers project from (**B**) the graft periphery toward the white matter of the (**C**) corpus callosum and traverse across the medial septum (**D**) into the (**E**) contralateral hemisphere. NeuN (red), Hoescht/DNA; Scale = 50 μm. (**F**–**K**) Dorsal striatal graft injection site 6 months following unilateral injection of iGABAs into the immunodeficient (**F**–**H**) R6/NSG HD and (**I**–**K**) WT/NSG mouse with insets demonstrating lateral and medial projection of human cytoplasm+ (green) fibers toward surrounding NeuN+ (red) neurons of the ipsilateral hemisphere. Scales, (**F**,**I**) = 500 μm, (**G**,**H**,**J**,**K**) = 50 μm.

**Figure 6 biology-14-00217-f006:**
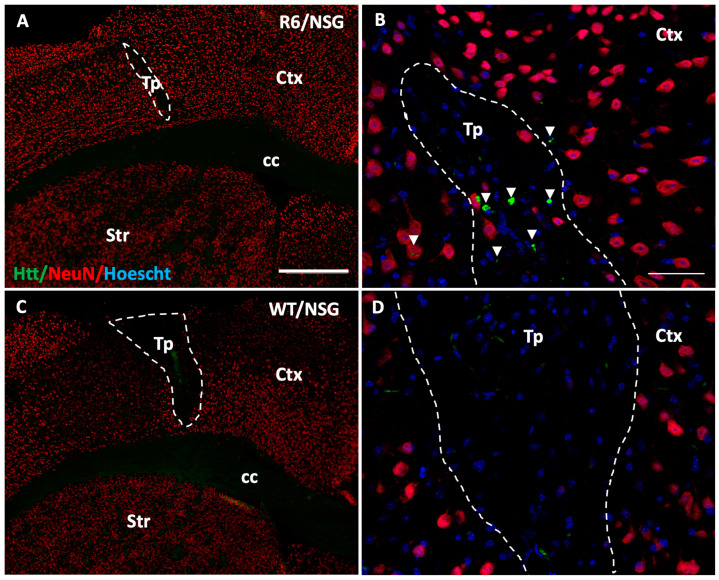
Infiltration of mutant htt into the iGABA engraftment site. (**A**–**D**) mHtt aggregates (green, arrow heads) were found in iGABAs grafts 6 months following transplantation in (**A**,**B**) R6/NSG HD mice but not (**C**,**D**) WT/NSG littermates. Dashed line indicates outside barriers of transplant as determined by presence of NeuN+ (red)/Hoescht+ (blue) cells. Abv: cc = corpus callosum, Ctx = cortex, Str = striatum, Tp = transplant. Scales, (**A**,**C**) = 500 μm, (**B**,**D**) = 50 μm.

**Table 1 biology-14-00217-t001:** Single-cell gene expression array primers.

ID	Forward Primer	Reverse Primer
GAPDH	GAACGGGAAGCTTGTCATCAA	ATCGCCCCACTTGATTTTGG
DLG4	AGCTGGAGCAGGAGTTCAC	ACACGCTTCACCTTGTGGTA
FOXG1	GCCAGCAGCACTTTGAGTTA	TGAGTCAACACGGAGCTGTA
DLX1	CATGACCACCATGCCAGAAA	CGGCCCAAACTCCATAAACA
DLX2	TTCGTCCCCAGCCAACAA	TGGCTTCCCGTTCACTATCC
SLC32A1	AGGCTGGAACGTGACCAAC	GAGAAACAACCCCAGGTAGCC
GAD2	CTGCTCCAAAGTGGATGTCAAC	AAAGTGGGCCTTTCTCCATCA
GAD1	ATCCTGGTTGACTGCAGAGAC	CCAGTGGAGAGCTGGTTGAA
SOX1	GGGTCAAACGGCCCATGAA	TTGTGCATCTTGGGGTTCTCC
CUX2	TCCATCACCAAGAGGGTGAA	CAGGATGCTTTCCCCAAACA
RELN	TCCAGAATTGGAAGCGGATCA	GTTGGCCTGAATCCATCTGAAC
SYN1	GCAAGGACGGAAGGGATCA	TGTCTTCATCCTGGTGGTCAC
HTR2C	AGACTGAAGCAATCATGGTGAAC	TGCTACTGGGCTCACAGAAA
PAX6	CCCCACATATGCAGACACACA	GAACTGACACACCAGGGGAAA
NR4A2	TGGCTGTTGGGATGGTCAAA	TCTTCGGTTTCGAGGGCAAA
FOXP2	AGGGACTCATCTCCATTCCA	GCTGAATCTCAGCAGGACTTA
POU3F2	CGGATCAAACTGGGATTTACCC	CGAGAACACGTTGCCATACA
BCL11B	CAACCCGCAGCACTTGTC	CCTCGTCTTCTTCGAGGATGG
ISL1	TCGCCTTGCAGAGTGACATA	CCCGGTCCTCCTTCTGAAAA
TBR1	ACGAACAACAAAGGAGCTTCA	TGGTACTTGTGCAAGGACTGTA
SLC17A6	TGGGGCTACATCATCACTCA	GAAGTATGGCAGCTCCGAAA
EMX2	GCCCCATAAATCCGTTCCTCA	CAAGTCCGGGTTGGAGTAGAC
LHX2	CAAAAGACGGGCCTCACCAA	CGTAAGAGGTTGCGCCTGAA
NEUROG1	CGACACCAAGCTCACCAAAA	CCAGAGCCCAGATGTAGTTGTA
ETV1	AGCCGTTCACTCCGCTATTA	AAGGGCTTCTGGATCACACA
ASCL1	TGGTGCGAATGGACTTTGGAA	CTCCCAACGCCACTGACAA
CALB2	TTGGCGGAAGTACGACACA	CCTTCTTCAGCAGGTCTGACA
SST	CCCAGACTCCGTCAGTTTCT	AGCAGCTCTGCCAAGAAGTA
SATB2	TTTGCCAAAGTGGCTGCAAA	TTTCTGGGCTTGGGTTCTCC
NKX2-1	GATGGTACGGCGCCAAC	CCATGCCGCTCATGTTCA

**Table 2 biology-14-00217-t002:** Subject recruitment.

Animal	Vendor	Number (Sex)	Sacrifice
Sprague Dawley Rat	Harlan Laboratories	3 (Female)	3 Months
RNU Nude	Charles River Laboratories	1 (Male)	7 Days
RNU Nude	Charles River Laboratories	5 (Male)	9 Months
R6/NSG WT	UC Davis Stem Cell Program	2 (Female)	3 Months
R6/NSG Tg	UC Davis Stem Cell Program	3 (Female)	3 Months
R6/NSG Tg	UC Davis Stem Cell Program	9 (Female)	6 Months
Cynomolgus Monkey	University of Illinois at Chicago	2 (Male)	1 Month

**Table 3 biology-14-00217-t003:** Antibody information.

Primary Antibody	Species	Company	Catalog#	Dilution	Assay
Calbindin (C26D12)	Rabbit	Cell Signaling	2173	1:200	IHC
DARPP32	Rabbit	Abcam	ab-40801	1:5000	IHC
Human Nuclei	Mouse	Millipore	MAB1281	1:400	IHC
Human NCAM (Eric-1)	Mouse	Santa Cruz	sc-106	1:1000	IHC
Human Cytoplasm	Mouse	Takara	SC121	1:1000	IHC
Ki-67	Rabbit	Abcam	ab-15580	1:500	IHC
mEM48	Mouse	Millipore-Sigma	MAB5374	1:500	IHC
NeuN	Rabbit	Abcam	ab177487	1:1000	IHC
Map2	Rabbit	Abcam	ab183830	1:1000	IHC
anti-Mouse AF-555	Donkey	Molecular Probes	A31570	1:500	IHC
anti-Rabbit AF-488	Donkey	Molecular Probes	A21206	1:500	IHC
anti-Rabbit AF-594	Goat	LifeTech	A32740	1:500	IHC
anti-Mouse AF-488	Goat	LifeTech	A32723	1:500	IHC

**Table 4 biology-14-00217-t004:** Stereological cell counts.

Animal #, Graft	Estimated Number of HuNuclei+ Cells	Coefficient of Error (Gundersen), m = 1	% Survival
#29, Left Caudate	116,505.62	0.06	25.89
#30, Left Caudate	217,355.33	0.04	48.30
#33, Left Caudate	956.19	0.16	0.21
#34, Left Caudate	165,083.25	0.08	36.69
#35, Left Caudate	97,738.29	0.10	21.72
#29, Right Caudate	129,043.14	0.06	28.68
#30, Right Caudate	123,517.76	0.10	27.45
#33, Right Caudate	116,236.52	0.08	25.83
#34, Right Caudate	156,574.52	0.05	34.79
#35, Right Caudate	109,047.51	0.07	24.23

## Data Availability

The data that support the findings of this study are available from the corresponding authors upon reasonable request.

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
