# Peer review of "Long-Term Engraftment of Cryopreserved Human Neurons for In Vivo Disease Modeling in Neurodegenerative Disease"

_biology, 2025, doi:10.3390/biology14020217_

Round 1
Reviewer 1 Report
Comments and Suggestions for Authors
The paper is very good. The study is well-structured and methodologically sound. The research is thorough.
The approach is very technical, but the cell culture techniques, immunohistochemistry and genetic analyses complement each other in a rigorous manner.
Concerning the approach, I was a little disturbed about the use of the monkeys, however it was a necessary effort, in order to expand the research to higher organism, even if this time it did not demonstrate any difference to the results in mice and rats. The references are rich and adequately placed.
A welcome addition could be a 2-3 paragraph exploration of the ethical dilemmas of xenografts or of long-term transplantation.
The readability is unavoidably low, as it is very technical and precise, it might present some challenges to non-specialist readers, however, it's a paper for the highly specialized. A more simplified approach could help, with bullet lists and a little less redundancy in the explanations.
Author Response
Reviewer 1: We kindly thank the Reviewer for the comments and recommendation to add a summary on xenotransplantation ethical concerns.
- A welcome addition could be a 2-3 paragraph exploration of the ethical dilemmas of xenografts or of long-term transplantation.
We agree and have now enriched the text with the following summary of the perceived ethical dilemmas regarding human brain xenografts in lower mammals.
“The ethical dilemmas surrounding human neural xenografts following long-term transplantation are multifaceted, touching on questions of identity, consciousness, and moral responsibility. One major concern is the potential for these transplants to integrate functionally into host brains in ways that could alter cognition, identity, or even subjective experience. According to the National Academies report on human neural organoids, transplants, and chimeras, such research raises the possibility of transplanted human neural tissue influencing host behavior, cognition, or self-awareness, particularly when integrated into nonhuman primates (53)​. This concern is amplified when considering long-term survival and adaptation of human neural cells in host brains, where they might mature beyond current expectations, leading to unpredictable cognitive or moral implications.
A related ethical challenge is the question of moral status. The presence of human neural tissue in another organism or in a significantly altered human brain challenges traditional frameworks used to determine moral worth and rights. Jeziorski et al. (54) discuss similar concerns in the context of brain organoids, particularly regarding the possibility of such entities developing forms of consciousness that warrant ethical consideration​. If human neural xenografts contribute to the formation of complex cognitive functions or subjective experiences, researchers and ethicists may need to reassess existing guidelines for their use, care, and potential termination. The long-term implications of such transplants necessitate rigorous oversight, public engagement, and ethical reflection to ensure that advances in neuroscience do not unintentionally create morally ambiguous entities or violate fundamental principles of autonomy and dignity.”
Reviewer 2 Report
Comments and Suggestions for Authors
Introduction; in line 63-65 “….have been widely reported…..” and line 69-71 “Extensive
evidence from numerous reports…..”, the references should be added.
Also in line 114 “….has been widely published”, the references should be added.
About “Cryopreserved neurons” in the present study, how long was the period of
cryopreserved? How bout the effects of the long of period?
3.2; how old of SD rats were used? The old of SD rats were not written in Materials and
Methods and in Results.
3.4; “at 1-minth post-injection” in line 304 and “at 3-months post-administration” in line 310,
which was correct?
Line 309, what was “NHP”?
The reference [31] in text was out of order.
The references [41], [42], and [43] were not present in text.
In Abbreviations, correct “Mod-Skid-Gamma” to “Nod-Skid-Gamma”.
Author Response
Reviewer 2: We kindly thank the Reviewer for the constructive comments and kind recommendations. We agree with all recommendations and have addressed all the suggested areas for improvement within the revised manuscript.
- Introduction; in line 63-65 “….have been widely reported…..” and line 69-71 “Extensive evidence from numerous reports…..”, the references should be added. Also in line 114 “….has been widely published”, the references should be added.
We thank the Reviewer for your diligence in proper citation. The appropriate references below were added in the respective spots and all other following references updated for accuracy and numerical order.
63-65: Svendsen SP, Svendsen CN. Cell therapy for neurological disorders. Nat Med. 2024;30(10): 2756-2770. doi: 10.1038/s41591-024-03281-3.
69-71: Guo JL, Lee VM. Cell-to-cell transmission of pathogenic proteins in neurodegenerative diseases. Nat Med. 2014;20(2):130-8. doi: 10.1038/nm.3457.
114: Okano H, Morimoto S. iPSC-based disease modeling and drug discovery in
cardinal neurodegenerative disorders. Cell Stem Cell. 2022;29(2):189-208. doi: 10.1016/j.stem.2022.01.007.
- About “Cryopreserved neurons” in the present study, how long was the period of
cryopreserved? How about the effects of the long of period?
We thank the reviewer for acknowledging the important critical quality attributes related to long-term batch stability of cryopreserved iGABA neurons. The lot of iGABA cryopreserved neurons used in the animal studies (Lot# 1956638) was cryopreserved and released in September 2014. For the Sprague Dawley rat, athymic rat, nonhuman primate, and mouse studies, the cryopreserved cells were thawed in March 2015, January 2015, March & April 2015, and April 2018 respectively. Long-term stability assessed by repeat-thaw analysis for up to 5 years has consistently shown product stability within release criteria for iGABA neurons. The manufacturer’s certificate of analysis guarantees quality following thaw within 1-year of the shipping date. The following text was added to the manuscript.
“Batch analysis of iGABA commercial lots consistently demonstrate stability of critical quality attribute release criteria up to 5 years post-cryopreservation, the longest time point assayed by the manufacturer.”
- How old of SD rats were used? The old of SD rats were not written in Materials and
Methods and in Results.
We thank the reviewer for their diligence. The detail was inadvertently omitted. The age of the Sprague-Dawley rats at the time of iGABA infusion was 5 months old. The age was added to the manuscript for clarity.
- “at 1-month post-injection” in line 304 and “at 3-months post-administration” in line 310, which was correct?
The correct timepoint is 1-month as correctly labeled in Table 2. We thank the reviewer for their diligence, now corrected in the manuscript text.
- Line 309, what was “NHP”?
“NHP” refers to “nonhuman primate” which was originally an abbreviation in a late draft of the manuscript but inadvertently not corrected when the abbreviation was removed from the manuscript. We thank the reviewer for their diligence, now corrected in the manuscript text to “Cynomolgus monkey.”
- The reference [31] in text was out of order.
We thank the reviewer for their diligence, now corrected in the manuscript text.
- The references [41], [42], and [43] were not present in text.
We thank the reviewer for their diligence, now corrected in the manuscript text.
- In Abbreviations, correct “Mod-Skid-Gamma” to “Nod-Skid-Gamma”.
We thank the reviewer for their diligence, now corrected in the manuscript text.